# The GeriPACT Initiative to Prevent All-Cause 30-Day Readmission in High Risk Elderly

**DOI:** 10.3390/geriatrics6010004

**Published:** 2021-01-06

**Authors:** James S. Powers, Lovely Abraham, Ralph Parker, Nkechi Azubike, Ralf Habermann

**Affiliations:** 1Tennessee Valley Healthcare System, Nashville, TN 37212, USA; lovely.abraham@va.gov (L.A.); ralph.parker@va.gov (R.P.); Nkechi.azubike@va.gov (N.A.); ralf.habermann@va.gov (R.H.); 2TVHS Geriatric Research Education and Clinical Center, Nashville, TN 37212, USA; 3Vanderbilt Center for Quality Aging, Vanderbilt University Medical Center, Nashville, TN 27203, USA

**Keywords:** transitions of care, 30-day hospital readmissions, geriatrics, patient-centered medical home, interdisciplinary team, quality improvement

## Abstract

Background: Suboptimal care transitions increases the risk of adverse events resulting from poor care coordination among providers and healthcare facilities. The National Transition of Care Coalition recommends shifting the discharge paradigm from discharge from the hospital, to transfer with continuous management. The patient centered medical home is a promising model, which improves care coordination and may reduce hospital readmissions. Methods: This is a quality improvement report, the geriatric patient-aligned care team (GeriPACT) at Tennessee Valley Healthcare System (TVHS) participated in ongoing quality improvement (Plan, Do, Study, Act (PDSA)) cycles during teamlet meetings. Post home discharge follow-up for GeriPACT patients was provided by proactive telehealth communication by the Registered Nurse (RN) care manager and nurse practitioner. Periodic operations data obtained from the Data and Statistical Services (DSS) coordinator informed the PDSA cycles and teamlet meetings. Results: at baseline (July 2018–June 2019) the 30-day all-cause readmission for GeriPACT was 21%. From July to December 2019, 30-day all-cause readmissions were 13%. From January to June 2020, 30-day all-cause readmissions were 15%. Conclusion: PDSA cycles with sharing of operations data during GeriPACT teamlet meetings and fostering a shared responsibility for managing high-risk patients contributes to improved outcomes in 30-day all-cause readmissions.

## 1. Introduction

### 1.1. Problem Description

The cost of unplanned all-cause 30-day hospital readmissions to the Medicare program is estimated at over $12 billion yearly [1]. Beyond economic implications, suboptimal care transitions increases the risk of adverse events resulting from poor care coordination among providers and healthcare facilities [2]. Repeat hospitalizations are tied to poor transitions of care and many existing barriers.

### 1.2. Available Knowledge

Suggestions for improvement have been advanced by the National Transitions of Care Coalition, a work group convened by the Patient-Centered Outcomes Research Institute (PCORI) identified eight transitional care components and operational strategies to enhance transitional care, based on real-world patient and caregiver experiences. The categories included (1) patient engagement, (2) caregiver engagement, (3) complexity/medication management, (4) patient/caregiver education and respecting caregiver role, (5) patient and caregiver well-being, (6) care continuity, (7) accountability of clinician, team, and caregiver, and (8) clinician and team organization [3]. These barriers have implications for enhanced clinical care models, including comprehensive assessment and person-centered care, monitoring progress, and providing caregiver resources, identifying high-risk patients, reducing polypharmacy, addressing health literacy, team communication, and enhanced collaboration with referral sources. The National Transition of Care Coalition recommends shifting the discharge paradigm from discharge from the hospital to transfer with continuous management. Practice changes that could potentially impact readmission rates include provider education regarding realistic post-acute care capabilities, emerging strategies to improve coordination, communication, and cooperation among healthcare professionals across healthcare settings, and maximizing stability of multiple comorbidities prior to discharge.

### 1.3. Rationale

The patient centered medical home is a promising model, which includes many of these components and may reduce hospital readmissions.

### 1.4. Aim

We describe our experience implementing this model with a goal of improving post-hospital transitions of care.

## 2. Materials and Methods 

### 2.1. Context

The Department of Veterans Affairs (VA) operates Tennessee Valley Healthcare System (TVHS), an integrated healthcare system of over 100,000 patients in Middle Tennessee comprised of 2 medical centers located 40 miles apart, and 12 community-based outpatient clinics. In 2011, TVHS developed a geriatric patient-centered medical home model for geriatric primary care—the geriatric patient-aligned care team (GeriPACT) [4,5]. The GeriPACT Team consists of the GeriPACT provider (geriatrician or geriatric nurse practitioner with an outpatient panel size of approximately 800), a social worker, a clinical pharmacist, a registered nurse care manager, a licensed vocational nurse, and clerical staff. GeriPACT is a special population PACT within primary care for complex geriatric and other high-risk vulnerable veterans providing integrated, interdisciplinary assessment and longitudinal management, and coordination of both VA-sponsored and non-VA sponsored (Medicare and Medicaid) services for patients and caregivers. GeriPACT at the Nashville Campus of TVHS has an enrollment of 745 patients with an average age of 84 years, taking an average of 8.7 medications, with 12% congestive heart failure, 10% chronic lung disease, 30% diabetes mellitus, 15% dementia, and a 12% yearly mortality. Some 48 patients receive chronic prescription opioid therapy. The practice is supported by the Department of Veterans Affairs Computerized Patients Record System (CPRS), including the electronic patient portal, *My* health*e*vet, with telemedicine capabilities. Data were collected by chart review with operations data extracted. 

Risk Stratification: the clinical assessment of need (CAN) is a clinical predictor of future hospitalization and death developed for VA populations [6]. This methodology extracts predictors from 6 categories: social demographics, medical conditions, vital signs, prior year use of health services, medications, and laboratory tests and constructs logistic regression models to predict outcomes. CAN scores are from 1 to 99, with higher scores corresponding to an increased probability of future healthcare events. Approximately 25% of this high-risk population with a mean CAN score of approximately 70 are hospitalized yearly, and most GeriPACT patients utilize the VA hospital for their care.

### 2.2. Intervention

In July 2019, GeriPACT at TVHS participated in rapid-cycle (RCQI) [7] ongoing quality improvement (Plan, Do, Study, Act, (PDSA)) cycles during teamlet meetings.

Quality Improvement (QI)Process: Plan, Do, Study, Act (PDSA) cycles, originally described by Deming of Bell Laboratories, were adopted to introduce quality improvement into primary care. Deciding on a topic important to the practice, implementing, and measuring change can help the teamwork collectively to achieve common goals. PDSA Cycles include the following components:Plan—plan the test for observation, including a plan for collecting data, state the objective of the test;Do—try out the test on a small scale, document problems, and observations;Study-check—analyze the data, compare the data to previous experience and reflect on what was learned;Act—refine the change based on what was learned from the test, determining what modification should be made and develop a plan for the next test.

Hospital case managers notified GeriPACT daily about VA Hospital discharges to home. Post home discharge follow-up for patients was provided by proactive telehealth communication (phone and video connect) by the RN care manager and nurse practitioner, preventing potential gaps in care by addressing medication reconciliation, referrals and follow-up appointments, goals of care, advance directive completion, home and community-based resources, and caregiver concerns. Approximately 6 open access visits per week were generated after contacting patients. Periodic operations data obtained from the Data and Statistical Services Coordinator informed the PDSA cycles and teamlet (treatment team) meetings.

### 2.3. Measures

The Merit-Based Incentive Payment System (MIPS), Medicare, Quality Measure 356 is defined as Measure Description—Numerator: total number of unique GeriPACT patients > age 65 with all-cause re-admissions to TVHS within 30 days of discharge during the measurement period. Denominator: total number of index hospitalizations that occurred during the measurement period (excludes repeat hospitalizations within a 30-day period) [8].

This is a quality improvement report and has been informed by the Standards for QUality Improvement Reporting Excellence (SQUIRE) criteria [9], and this report meets the quality improvement minimum quality criteria set (QI-MQCS) domains for reporting quality improvement work. [10]. The TVHS Institutional Review Board (IRB) has determined this study as a quality improvement initiative.

## 3. Results

Results include pre- and post-intervention data of numbers of unique individuals over age 65 having a repeat hospitalization at TVHS within 30 days, divided by the total number of index hospitalizations (excludes repeat hospitalizations within a 30-day period). At baseline (July 2018–June 2019) the 30-day all-cause readmission for GeriPACT was 41/191 (23%). From July to December 2019, 30-day all-cause readmissions were 13/101 (18%). From January to June 2020, 30-day all-cause readmissions were 12/79 (15%). 

## 4. Discussion

A randomized trial including 3054 patients showed no evidence of an impact on 30-day readmissions from a hospital-managed nurse follow-up telephone call program within 3 to 7 days of discharge, designed to assess understanding and provide education and assistance to support discharge plan implementation, but not provide ongoing continuity of care [11]. GeriPACT provides continuity of care and advanced open access scheduling to accommodate patient and caregiver needs. GeriPACT functions as an autonomy supportive model promoting a participatory group function, which encourages members to function at the top of their scope of practice [12]. This form of team management promotes self-realization of individual team members and positive reinforcement in a supportive, non-competitive environment. Sharing of operations data and strategic planning for complex care management require active teamlet input and promote sharing of outcomes. Implementation challenges included the turnover of several individual team members during the intervention, determining follow-up strategies between the nurse practitioner and the nurse care manager relative to individual patient complexity, and timely consultations with other team members. Devoting time during busy teamlet meetings for quality improvement activities necessitated scheduling due to competing collateral duties of individual team members and frequent patient management discussions for individual complex-care patients.

Limitations: this study was conducted at a Veterans Health Administration facility and conclusions may not be directly applicable to other settings. However, our results may encourage development of patient centered medical homes in other healthcare systems. While all patients utilized the VA as a primary source of care, we could not account for potential patients admitted to community hospitals. Only home discharges from the VA hospital were included in the intervention. Our project overlapped the COVID (Coronovirus, SARS-COV-2) pandemic, and while this did not influence the telehealth intervention, we cannot determine its effect on patient decisions to seek hospital care.

## 5. Summary

PDSA cycles with sharing of operations data during GeriPACT teamlet meetings and fostering a shared responsibility for managing high-risk patients contributes to improved outcomes in 30-day all-cause readmissions.

## Data Availability

Data available upon request from the corresponding author (JP).

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
