# Peer review of "The GeriPACT Initiative to Prevent All-Cause 30-Day Readmission in High Risk Elderly"

_geriatrics, 2021, doi:10.3390/geriatrics6010004_

Round 1

Reviewer 1 Report

Summary: This paper describes the GeriPACT transitions of care program at the Tennessee Valley Healthcare System (Nashville) & Veterans Administration program. This program focuses on high risk veterans being discharged to home. The GeriPACT program involves a multidisciplinary team who provides support in the postdischarge period to help improve care and reduce readmissions. These kind of interventions are absolutely key to reducing readmissions which is a high priority for our national health system, individual health systems, hospitals, payers as well as patients. Quality improvement projects and articles like this are absolutely essential to encourage so that we can actualize on our hopes of improving transitions of care reducing readmissions.

Broad comments:

Strengths: 1. A major strength is the fact that this paper describes a highly functioning and integrated multidisciplinary team focused on transitions of care. The team used communications strategies, PDSA cycles, and team lead meetings to do their work. This is not easy to pull off and required major effort & dedication of multiple individuals to move its forward.

  1. Readmissions rates did decrease from the baseline to the postintervention period in a basic pre- post- analysis.

Opportunities:

  1. Study definition and results – the authors indicate their framework of following Squire reporting criteria article. I do believe the section around study design definition and results should be reviewed and elaborated on. This is a fairly standard quality improvement project with pre— post – measurement of readmissions that were affected (presumably) by the GeriPACT program intervention. The results that are presented are total readmission numbers. A few thoughts on this:
    1. The authors should consider defining how readmissions were measured & reported– is this the CMS definition or an internal health system or VA definition? Is this readmission to all hospitals? Single hospital? Health system hospitals?
    2. Statistical analysis in articles like this is very tricky to do which is why understandably none was done or reported by the authors. However certain other data points or visual presentation of the data may be helpful to the reader to understand whether the reduction readmissions could be attributed to the GeriPACT program. I would suggest:
      1. Adding a table 1 which includes a description of the population that is being intervening on – demographics, CAN scores, whatever other demographic or clinical data in the data set for the GeriPACT program. The number of patients on chronic opiates was provided, what about other clinical descriptions which may be of interest to the reader?
      2. Adding the number of patients that were intervened on (numerator and denominator).
      3. Consider a time-based control run chart of 30 day readmission rates by month for the period of January 2019- June 2020. This may be helpful in presenting the case for correlation of the program intervention to readmissions reductions as you would most likely see month by month readmission rates dropped after the program starts.
    3. Intervention description: I believe the reader can get a basic understanding of the intervention that occurred in the GeriPACT program but I think some more detail would be helpful. How many visits on average did each patient get after discharge? How many visits in person versus telehealth? Since your measurement period included the COVID pandemic era how did COVID affect the visits? Did the intervention team do any work before hospital discharge? The discussion mentions advanced open access scheduling which I think the readers would be very interested to hear more detail about that if that’s a significant part of the intervention. Was a template or checklist used for each patient to track the interventions? How were goals of care discussions facilitated – palliative care? Hospice referrals? I think these are some of the questions that the average reader will have when they read this article and try and adopt the lessons learned to perhaps their own practices.
    4. COVID: in our health system we have experienced significant reductions in thirty-day readmission rates which we would love to attribute to our interventions, however, we suspect that COVID has installed the fear in patients and their families are coming back to the hospital and that is probably what is driving reduction in admissions & readmissions nationally. This has also been reported publically. It is probably worth mentioning this in your limitations as it’s hard to untangle the COVID pandemic effects on admissions/readmissions reduction with the results on QI interventions like this.

Specific Comments:

  1. Line 21: the word “teamlet” is used frequently throughout the article. I think I know what that means but it might be helpful to give a definition.
  2. Line 37: Need reference for cost of readmissions estimate
  3. Line 39: Also suggest reference for negative impact of readmissions
  4. Line 51-52: To make this read more clear I wonder about either putting in quotes (or bolding) “discharge from the hospital”, to “transfer with continuous management”
  5. Line 60-62: I got a little confused in reading that this was supported by Tennessee Valley healthcare system yet was also VA patients. These are separate systems. Can you add any clarity as to what TVHS is responsible for or provided versus the VA?

Thanks very much to the teams at TVHS & Nashville VA for doing this important work and I congratulate them! Papers like these are essential to get published so that other health providers & systems can learn and adopt similar strategies. The edits & suggestions I have made are in the spirit so that the paper can be more accepted and understood by general readers. Thanks for the opportunity to do this review

Author Response

Thank you for your review of our manuscript which is revised and attached below.

1. Study methodology, definitions, and results have been revised with additional detail, reference to CMS MIPS Measure for unplanned re-hospitalizations, and definitions, and data about impact on open access added.

Hospital case managers notified GeriPACT daily about VA Hospital discharges to home. Post home discharge follow-up for patients was provided by proactive telehealth communication (phone and video connect) by the RN Care Manager and Nurse Practitioner, preventing potential gaps in care by addressing medication reconciliation, referrals and follow-up appointments, goals of care, advance directive completion, home and community-based resources, and caregiver concerns.  Approximately 6 open access visits per week were generated after contacting patients. Periodic operations data obtained from the Data and Statistical Services Coordinator informed the PDSA cycles and teamlet (treatment team) meetings.

MIPS Quality Measure 356 is defined as Measure Description: Numerator:  Total number of unique GeriPACT patients > age 65 with all-cause re-admissions to TVHS within 30 days of discharge during the measurement period. Denominator: Total number of index hospitalizations that occurred during the measurement period (excludes repeat hospitalizations within a 30-day period). [6]

[6] MIPS Quality Measure 356: Unplanned all-cause hospital readmission. https://qpp.cms.gov/docs/QPP_quality_measure_specifications/CQM-Measures/2019_Measure_356_MIPSCQM.pdf Accessed 12/8/20.

2. Results have been expanded to include revised numerators and denominators as reflected in the definitions above.

Results

Results include pre and post intervention data of numbers of unique individuals over age 65 having a repeat hospitalization at TVHS within 30 days divided by the total number of index hospitalizations (excludes repeat hospitalizations within a 30-day period). At baseline (July 2018-June 2019) the 30-day all cause readmission for GeriPACT was 41/191 (23%). From July-Dec 2019, 30-day all-cause readmissions were 13/101 (18%). From Jan-June 2020, 30-day all-cause readmissions were 12/79 (15%).

3. The GeriPACT population is defined with more data given within the text to maintain brevity for this short QI communication.

GeriPACT at the Nashville Campus of TVHS has an enrollment of 745 patients with an average age of 84 years, taking an average of 8.7 medications, with 12% congestive heart failure, 10% chronic lung disease, 30% diabetes mellitus, 15% dementia, and a 12% yearly mortality. Some 48 patients receive chronic prescription opioid therapy. The practice is supported by the Department of Veterans Affairs Computerized Patients Record System (CPRS), including the electronic patient portal, My healthevet, with telemedicine capabilities. Data were collected by chart review with operations data extracted.

Risk Stratification: The clinical assessment of need (CAN) is a clinical predictor of future hospitalization and death developed for VA populations. [4] This methodology extracts predictors from 6 categories: social demographics, medical conditions, vital signs, prior year use of health services, medications, and laboratory tests and constructs logistic regression models to predict outcomes. CAN scores are from 1-99, with higher scores corresponding to an increased probability of future healthcare events. Approximately 25% of this hi-risk population with a mean CAN score of approximately 70 are hospitalized yearly, and most GeriPACT patients utilize the VA hospital for their care.

4. Limitations section is expanded to include reference to COVID and to detail lack of ability to account for potential admissions to community hospitals.

Limitations: This study was conducted at a Veterans Health Administration facility and conclusions may not be directly applicable to other settings. However, our results may encourage development of patient centered medical homes in other healthcare systems. While all patients utilized the VA as a primary source of care, we could not account for potential patients admitted to community hospitals. Only home discharges from the VA hospital were included in the intervention. Our project overlapped the COVID pandemic, and while this did not influence the telehealth intervention, we cannot determine its effect on patient decisions to seek hospital care.

5.  Data runs were set for 6 month intervals with our quality management team  in this ongoing QI program as opposed to shorter time frames due to the need to adjust for the 30 day interval for unplanned rehospitalizations.

6. References for the costs and negative impacts of unplanned rehospitalizations are added:

Medicare Payment Advisory Commission. A path to bundled payment around a rehospitalization. In: Report to the Congress: Reforming the delivery system. Washington, DC, June 2005:83-103.

Naylor MD, Shaid EC, Carpenter D et al. Components of comprehensive and effective transitional care. J Am Geriatr Soc. 2017;65:1119–1125.

7.  Clarification of VHA and TVH is explained:

The Department of Veterans Affairs operates Tennessee Valley Healthcare System (TVHS), an integrated health care system of over 100,000 patients in Middle Tennessee

8. Transfer with continuous management is clarified by italics:

The National Transition of Care Coalition recommends shifting the discharge paradigm from discharge from the hospital, to transfer with continuous management

Reviewer 2 Report

This paper does not merit publication.  

I would recommend that the authors model their manuscript after other published work to help in using the proper format and language of a manuscript. 

The results section is only 2 sentences long. 

The objectives of the work are not clearly stated. 

Author Response

Thank you for your review of our manuscript. A revision is attached below which includes responses to all reviewers.

1. We agree the manuscript is better suited as a Communication due to the QI nature of our intervention and have changed the title to reflect this:

Communication

The GeriPACT Experience: Preventing All-cause 30-day Readmission in High Risk Elderly

James S. Powers MD* 1,2,3, Lovely Abraham ANP1, Ralf Parker RN1, Nkechi Azubike ANP1, Ralf Habermann MD1

2. The objective of the project are clarified in the introduction:

The patient centered medical home is a promising model which includes many of these components and may reduce hospital readmissions. We describe our experience implementing this model with a goal of improving post-hospital transitions of care.

3. The results section is expanded with revised data and definitions:

Results

Results include pre and post intervention data of numbers of unique individuals over age 65 having a repeat hospitalization at TVHS within 30 days divided by the total number of index hospitalizations (excludes repeat hospitalizations within a 30-day period). At baseline (July 2018-June 2019) the 30-day all cause readmission for GeriPACT was 41/191 (23%). From July-Dec 2019, 30-day all-cause readmissions were 13/101 (18%). From Jan-June 2020, 30-day all-cause readmissions were 12/79 (15%).